# Learning Efficient Vision Transformers via Fine-Grained Manifold Distillation

**Zhiwei Hao[1,2], Jianyuan Guo[2], Ding Jia[2,3], Kai Han[2], Yehui Tang[2,3],**
**Chao Zhang[3], Han Hu[1*], Yunhe Wang[2*]**

[1]School of information and Electronics, Beijing Institute of Technology.
[2]Huawei Noah's Ark Lab.
[3]Key Laboratory of Machine Perception (MOE),
School of Intelligence Science and Technology, Peking University.

{haozhw, hhu}@bit.edu.cn, {jianyuan.guo, kai.han, yunhe.wang}@huawei.com,
jiading@stu.pku.edu.cn, {yhtang, c.zhang}@pku.edu.cn

## Abstract

In the past few years, transformers have achieved promising performance on various computer vision tasks. Unfortunately, the immense inference overhead of most existing vision transformers withholds them from being deployed on edge devices such as cell phones and smart watches. Knowledge distillation is a widely used paradigm for compressing cumbersome architectures into compact students via transferring information. However, most of them are designed for convolutional neural networks (CNNs), which do not fully investigate the character of vision transformers. In this paper, we fully utilize the patch-level information and propose a fine-grained manifold distillation method for transformer-based networks. Specifically, we train a tiny student model to match a pre-trained teacher model in the patch-level manifold space. Then, we decouple the manifold matching loss into three terms with careful design to further reduce the computational costs for the patch relationship. Equipped with the proposed method, a DeiT-Tiny model containing 5M parameters achieves 76.5% top-1 accuracy on ImageNet-1k, which is +2.0% higher than previous distillation approaches. Transfer learning results on other classification benchmarks and downstream vision tasks also demonstrate the superiority of our method over the state-of-the-art algorithms.

## 1 Introduction

The past decade has witnessed the rise of attention-based models in the field of natural language processing (NLP) [1, 2]. Such models belonging to the transformer family [3] can effortlessly build long-range dependencies and have achieved remarkable performance. Inspired by the success in NLP, researchers have made great efforts introducing the transformer-based architectures to vision domain and achieved promising results. In an early attempt, Dosovitskiy *et al.* [4] proposed a transformer-based vision model termed ViT, which takes split image patches as the input. ViT obtains comparable performance when compared to the state-of-the-art convolutional neural networks (CNNs), demonstrating the immense potential of applying transformers to vision tasks. Inspired by the design of splitting the whole image into patches as input, various vision transformers have been proposed, including Swin [5], T2T [6], Twins [7], and TNT [8].

Although transformers have shown the excellent capability for various vision tasks, they require large amounts of parameters, resulting in heavy computational burden. For example, ViT-B [4] with 86M parameters pretrained on JFT-300M can only achieve comparable performance with CNN-based EfficientNet [9], while the latter is trained on ImageNet and contains only 5M parameters.

---

[*]Corresponding author.

36th Conference on Neural Information Processing Systems (NeurIPS 2022).

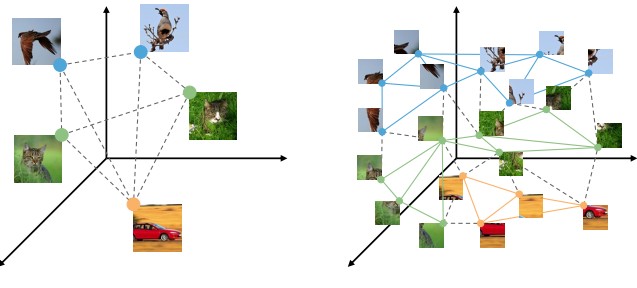

(a) Image-level manifold space     (b) Patch-level manifold space

Figure 1: Comparison between (a) image-level manifold space and (b) patch-level manifold space. The patch-level manifold space containing fine-grained information facilitates knowledge transfer. The requirement for inordinate computing resource and storage prevents them from being deployed on memory-bounded edge devices such as cell phones and smart watches. To facilitate the challenges above, a series of methods have been proposed to investigate compact deep neural networks, such as network pruning [10, 11], low-bit quantization [12, 13] and knowledge distillation [14, 15].

However, smaller models usually lead to performance degradation. Knowledge distillation (KD) [14] is a promising approach for inheriting information from a high-performance teacher to a compact student and maintaining the strong performance.

Touvron *et al.* [16] first proposed a KD-based compression approach for vision transformers. They trained the student transformer to match hard labels provided by a pre-trained CNN teacher. Although this approach obtains satisfying results, they ignore the intermediate-layer information inherited in vision transformers. There have been works [17, 18] proving the effectiveness of learning from intermediate layers for transformers in NLP, but these methods require teachers and students to have exactly the same embedding dimension at corresponding layers, which is a fairly tight constraint and cannot always be satisfied. Manifold learning-based KD [19, 20] can support layers with mismatching dimensions and make use of inter-sample information concurrently. However, existing manifold distillation approaches are designed for CNNs and cannot utilize the patch-level information of vision transformers. As shown in Figure 1, patches depict a manifold space in a more fine-grained way. Such information can facilitate knowledge transfer remarkably.

Based on this consideration, we propose a fine-grained manifold distillation method in both patch-level and batch-level. In particular, we regard vision transformers as projectors mapping inputs into multiple manifold spaces layer by layer. At each layer, we collect embeddings of patches to build their manifold relation map and train the student to match the relation map of the teacher. Since the computational complexity is high, we decouple the relation maps into three parts, which reduces the complexity by two orders of magnitude approximately. We evaluate the proposed method on the ImageNet-1k classification task. The proposed manifold KD outperforms the distillation method in [16] by +2.0% top-1 accuracy on DeiT-Tiny. We also conduct transfer learning experiments on CIFAR-10/100 and evaluate our method on downstream tasks such as object detection and semantic segmentation. The proposed method outperforms its counterparts on both tasks. Our contributions are summarized as follows:

- We propose a fine-grained manifold distillation method, which transfers patch-level and batch-level manifold information between vision transformers.
- We use three decoupled terms to describe the manifold space and simplify the computational complexity significantly.
- We conduct extensive experiments to verify the effectiveness of the proposed method. The results also demonstrate the importance of soft-label distillation and fixed-depth students.

## 2 Related works

**Vision transformer.** Transformer is originally designed for NLP tasks [21]. Inspired by its remarkable performance, researchers have made efforts to adopt transformer-based models in CV tasks [22, 23]. Among them, Dosovitskiy *et al.* [4] proposed to divide an image into patches and used embeddings of the patches as model input. Based on their proposed input processing scheme for images, many variants of vision transformers have been proposed. For example, Han *et al.* [8] proposed TNT to model in-patch attention, Zhou *et al.* [24] and Touvron *et al.* [25] built deeper vision transformers, and some researchers [26, 5, 27] adopted the experience in CNN to guide the design of vision

transformers. However, Most well-performed vision transformers are extremely resource-consuming and should be compressed for deployment.

**Knowledge distillation.** KD is a model compression method proposed by Hinton *et al.* [14], which trains a lightweight student model to match soft labels given by a large pre-trained teacher model [28]. Moreover, there are also works using structure of a information flow in the teacher model as knowledge [29, 30, 31], combining self-supervised training with KD [32], or training a student with only a few samples [11]. Due to its excellent performance, KD has been adopted in various research fields, such as CV [11, 32], NLP [17], and recommendation systems [33]. To compress vision transformers, Touvron *et al.* [16] proposed a KD-based method termed DeiT. They added a distillation token into the student and trained the student with hard labels provided by the teacher. Although DeiT has achieved remarkable performance, KD for vision transformers has not been well explored yet.

**Manifold learning.** Manifold learning is an approach for non-linear dimensionality reduction. It learns a smooth manifold embedded in the original feature space to construct low-dimensional features [34, 35]. Recently, several works introduce manifold learning to KD [19, 20]. These methods train the student to preserve the relationships among samples learned by the teacher. For vision transformers, these primary attempts are coarse and can be further improved because the basic input elements are patches not images.

## 3  Method

### 3.1  Preliminaries

**Vision transformer.** Vision transformers are attention-based models inherited from NLP, and each layer of transformer consists of a multi-head self-attention (MSA) block and a multi-layer perceptron (MLP) block. These models take split images as input, *i.e.*, the patches. In particular, assuming the patch is of size $P \times P$, an input image $\mathbf{x} \in \mathbb{R}^{H \times W \times C}$ of size $H \times W$ and channel number $C$ is reshaped into flattened patches $\mathbf{x_P} \in \mathbb{R}^{N \times (P^2 \cdot C)}$ for processing, where $N = HW/P^2$. The patches are then projected into a $D$-dimension embedding space and added to positional embeddings. We denote the summation result as $\mathbf{x}_e \in \mathbb{R}^{N \times D}$. Each model layer works as follows:

$$\mathbf{x}_e \leftarrow \text{MSA}(\text{LN}(\mathbf{x}_e)) + \mathbf{x}_e, \tag{1}$$
$$\mathbf{x}_e \leftarrow \text{MLP}(\text{LN}(\mathbf{x}_e)) + \mathbf{x}_e, \tag{2}$$

where LN denotes the layer normalization operation.

**Knowledge Distillation.** KD is a widely used model compression method, where we use predictions of a large pre-trained teacher model as the learning target of a tiny student model. Given a sample $\mathbf{x}$ corresponding to a label $y$, representing the prediction of the student and the teacher as $f_s(\mathbf{x})$ and $f_t(\mathbf{x})$, respectively, the loss function of KD can be formulated as:

$$\mathcal{L}_{kd} = (1 - \lambda)\mathcal{H}_{CE}(f_s(\mathbf{x}), y) + \lambda\tau^2\mathcal{H}_{KL}(f_s(\mathbf{x})/\tau, f_t(\mathbf{x})/\tau), \tag{3}$$

where $\mathcal{H}_{CE}$ is the cross-entropy function, $\mathcal{H}_{KL}$ is the Kullback-Leibler divergence function, $\tau$ is a label smoothing hyper-parameter termed temperature, and $\lambda$ is a balancing hyper-parameter.

Sometimes the intermediate features of the teacher can also be used for knowledge transfer. For example, Romero *et al.* [36] trained the student to output features similar to the teacher at intermediate layers. However, such methods require the teacher and the student to have the same embedding dimension. Otherwise, additional mapping layers are required for aligning, making the distillation process non-transparent. Manifold learning-based KD methods [20, 19] support mismatched embedding dimensions. They train the student to learn sample relationships predicted by the teacher but ignore patch-level information in vision transformers.

### 3.2  Fine-grained manifold distillation

To utilize the batch-level and patch-level information, we propose a fine-grained manifold distillation method. Unlike existing KD methods [16] for vision transformer only distilling with logits, our method distills batch and patch level manifolds at intermediate layers. Figure 2 provides an overview of the proposed method.

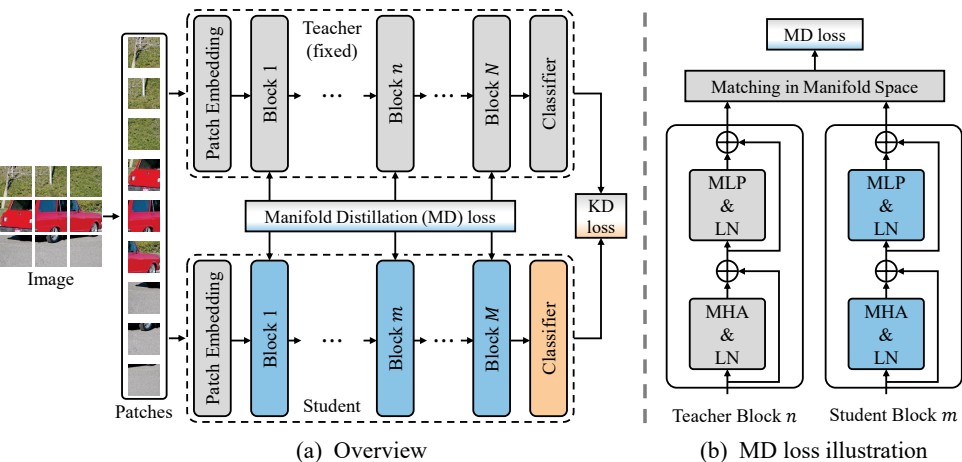

| (a) Overview | (b) MD loss illustration |

Figure 2: The fine-grained manifold distillation method. (a) An overview of the method. When transferring knowledge from the teacher to the student, a manifold distillation loss is adopted together with the orignal KD loss. (b) Computing of the manifold distillation loss. The loss is computed by matching feature relationships between each pair of selected teacher-student layers in manifold space.

In the fine-grained manifold distillation method, we regard a vision transformer as a feature projector that embeds image patches into a series of smooth manifold space layer by layer. At each pair of manually selected teacher-student layers, we aim to teach the student layer to output features having the same patch-level manifold structure as the teacher layer. In particular, given samples of batch size $B$, we denote the feature of the student layer and the teacher layer as $F_S \in \mathbb{R}^{B \times N \times D_S}$ and $F_T \in \mathbb{R}^{B \times N \times D_T}$, respectively, where $D_S$ and $D_T$ are embedding dimensions. We first normalize the feature at the last dimension and then compute the manifold structure, or manifold relation map, as follows:

$$\mathcal{M}(\psi(F_S)) = \psi(F_S)\psi(F_S)^T, \tag{4}$$

where $\psi : \mathbb{R}^{D_1 \times D_2 \times D_3} \to \mathbb{R}^{D_1 D_2 \times D_3}$ is a tensor reshape operation. The manifold relation map of $F_T$ can be obtained in a similar way. After that, we train the student to minimize the gap between $\mathcal{M}(\psi(F_T))$ and $\mathcal{M}(\psi(F_S))$ with the following loss:

$$\mathcal{L}_{mf} = \|\mathcal{M}(\psi(F_S)) - \mathcal{M}(\psi(F_T))\|_F^2. \tag{5}$$

However, the computation of manifold relation maps is resource-consuming. Its computational complexity is $\mathcal{O}(B^2 N^2 D)$ and a memory space of $B^2 N^2$ size is required to save the result. Taking the settings $B = 128$, $N = 196$ and $D = 192$ in the DeiT-Tiny model [16] as an example, it requires more than 240GFLOPs to compute and 2.5GB of memory space to save a single manifold relation map. The remarkable resource consumption limits the fine-grained manifold distillation method scaling up to multiple layers. Hence we must simplify the computation.

Inspired by the orthogonal decomposition of matrices, we decouple a manifold relation map into three parts: an intra-image relation map, an inter-image relation map, and a randomly sampled relation map. Figure 3 illustrates the decoupling. We compute the intra-image patch-level manifold distillation loss as follows:

$$\mathcal{L}_{intra} = \frac{1}{B} \sum_{i=0}^{B} \|\mathcal{M}(F_S[i,:,:]) - \mathcal{M}(F_T[i,:,:])\|_F^2. \tag{6}$$

Similarly, the inter-image patch-level manifold distillation loss is computed by:

$$\mathcal{L}_{inter} = \frac{1}{N} \sum_{j=0}^{N} \|\mathcal{M}(F_S[:,j,:]) - \mathcal{M}(F_T[:,j,:])\|_F^2. \tag{7}$$

Moreover, to relieve the information loss caused by the decoupling, we relate the intra-image and the inter-image manifolds via a relation map computed across randomly sampled patches. Specifically, we sample $K$ rows in the reshaped feature $\psi(F)$ to obtain $F^r \in \mathbb{R}^{K \times D}$, and compute the randomly sampled patch-level manifold distillation loss as follows:

$$\mathcal{L}_{random} = \|\mathcal{M}(F_S^r) - \mathcal{M}(F_T^r)\|_F^2. \tag{8}$$

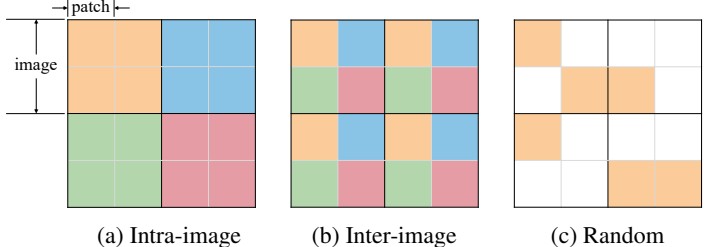

| (a) Intra-image | (b) Inter-image | (c) Random |

Figure 3: Illustration of the decoupled manifold relation map (4 images with 4 patches in each image): (a) intra-image relation map; (b) inter-image relation map; and (c) randomly sampled relation map. Manifold relation maps are computed across each group of patches filled with the same color.

Hence, the overall loss function of our proposed method is:

$$\mathcal{L} = \mathcal{L}_{kd} + \sum_l \mathcal{L}_{mf-decouple}, \tag{9}$$

$$\mathcal{L}_{mf-decouple} = \alpha \mathcal{L}_{intra} + \beta \mathcal{L}_{inter} + \gamma \mathcal{L}_{random}, \tag{10}$$

where $\alpha$, $\beta$, and $\gamma$ are hyper-parameters. The summation over $l$ means that the manifold relation map matching is performed on multiple pairs of teacher-student layers. We will discuss the layer selecting scheme in Section 4.3.

**Complexity analysis.** The computational complexity of the decoupled manifold relation map is $\mathcal{O}(BN^2D + B^2ND + K^2D)$, and the memory space requirement is $BN^2 + NB^2 + K^2$, where each one is reduced by nearly $BN/(B+N)$ times when ignoring the lower order terms. Still taking DeiT-Tiny as an example, the floating-point operations and memory space requirement reduce to 3GFLOPs and 32MB, respectively. The sampling number $K$ is set to 192, the same as our experiments. With the decoupling scheme, fine-grained manifold distillation across multiple layers becomes feasible.

### 3.3 Optimization

**Patch merging.** Although the decoupling reduces the computation and memory space remarkably, the computing and storing overhead is still unaffordable when the patch size is too small. For instance, the patch size at the first stage of SwinTransformer [5] is 4, indicating that the total patch number $N$ is 3136 under the input size of 224×224. Such a large patch number significantly increases the computational complexity and memory space requirement of the intra-image patch-level manifold loss $\mathcal{L}_{intra}$. To remedy this drawback, we merge adjacent patches and view them as a single patch to further simplify the computation. In particular, given a feature map $F \in \mathbb{R}^{N \times D}$, we first reshape it into $F^r \in \mathbb{R}^{H \times W \times D}$, where the height $H$ and the width $W$ are obtained following the original patch splitting operation. Then we adopt a merging setting $(H', W')$ to merge every $(H/H') \times (W/W')$ adjacent patches in a non-overlapping manner. Zero-padding is adopted when $H/H'$ or $W/W'$ is not an integer. After merging, the feature map becomes $F^{rm} \in \mathbb{R}^{H' \times W' \times (HWD/H'W')}$ and is finally reshaped into $F^m \in \mathbb{R}^{(H'W') \times (HWD/H'W')}$. By adjusting the merging setting $(H', W')$, we can easily strike the trade-off between the complexity and the granularity of manifold relation maps.

**Soft distillation.** Previous work [16] adds an additional distillation token into the student, and uses this token to learn hard labels provided by a CNN teacher. However, when teacher and student are both vision transformers, we propose that it is better to teach the student only with soft labels of the teacher. This design is based on the assumption that a larger model can learn more knowledge than a smaller one, and models of the same family share the same knowledge pattern, *i.e.*, a student can learn most knowledge of a teacher. In our work, unless the student is larger than the teacher, we set the hyper-parameter $\lambda$ in Equation 3 to 1.

**Fixed depth.** Stochastic depth [37] is a regularization method that has becomes an infrastructure in training vision transformers [16, 5]. We propose that the student should not adopt this regularization when the teacher is trained with stochastic depth, since the soft labels already contain knowledge about this regularization. Otherwise, the repeated regularizations may harm the student performance. Hence, we adopt a fix-depth student in our method, *i.e.*, the stochastic depth regularization is not used for training the student.

Table 1: Details of used teacher and student models. The input resolution is set to $224\times224$.

| Model | Embedding | Heads | Layers | #params | FLOPs | Throughput(im/s) |
|-------|-----------|-------|--------|---------|-------|------------------|
| RegNetY-16GF [38] | - | - | - | 84M | 15.9G | 334.7 |
| CaiT-S24 [25] | 384 | 8 | 24 | 47M | 9.4G | 573.6 |
| CaiT-XXS24 [25] | 192 | 4 | 24 | 12M | 2.5G | 1012.8 |
| DeiT-Small [16] | 384 | 6 | 12 | 22M | 4.6G | 940.4 |
| DeiT-Tiny [16] | 192 | 3 | 12 | 5M | 1.3G | 2536.5 |
| Swin-Small [5] | 96 | 3 | 24 | 50M | 8.7G | 436.9 |
| Swin-Tiny [5] | 96 | 3 | 12 | 29M | 4.5G | 755.2 |

## 4 Experiments

We evaluate our fine-grained manifold distillation method on ImageNet-1k [39] classification task, CIFAR-10/100 [40] transfer learning task, COCO [41] object detection task, and ADE20K [42] semantic segmentation task. Our implementation is based on Pytorch framework [43] and the MindSpore [2] Lite tool [44].

### 4.1 Setup

**Datasets.** We evaluate the proposed method mainly on the ImageNet-1k image classification task. ImageNet-1k is a subset of the ImageNet dataset [39], which consists of more than 1.2M training images and 50K validation images from 1000 classes. we conduct transfer learning experiments on CIFAR-10 and CIFAR-100 datasets [40] to test the generalization performance of student models. These two datasets both contain 50K training images and 10K testing images, which are categorized into 10 classes and 100 classes, respectively. Moreover, we conduct experiments on the object detection downstream task with COCO dataset [41]. We use the COCO 2017 split, which consists of 118K training images and 5K validation images containing objects from 80 categories.

**Baselines and models.** We compare the proposed method with DeiT [16], which adds an additional distillation token in the student model to learn hard labels from the teacher. SwinTransformer students containing no distillation token, so we compare with the original KD method [14] on these models. Table 1 summarizes the used models in our experiments.

### 4.2 Distillation results on ImageNet-1k

**Implementation details.** On the ImageNet-1k image classification task, we train DeiT students with CaiT teachers and SwinTransformer students with SwinTransformer teachers. We slightly modify the architecture of DeiT students by removing the distillation token and only using the class token. The hyper-parameter $\lambda$ in the KD loss is set to 1, *i.e.*, the real label is not used to train the student. When the teacher is smaller than the student, to prevent the performance degradation caused by the weak teacher, we set $\lambda$ to 0.5. In the fine-grained manifold distillation loss, hyper-parameters $\alpha$, $\beta$, and $\gamma$ are set to 4, 0.1, and 0.2, respectively. The sampling number $K$ in loss term $\mathcal{L}_{random}$ is set to 192. We set the above hyper-parameters *one by one* with the grid search method, indicating that their combination may not be optimal. We select the first 4 layers and the last 4 layers of the student and the teacher to conduct manifold distillation. Note that the class token in DeiT is ignored when computing manifold relation maps. Moreover, to train SwinTransformer students efficiently, we adopt a patch merging setting of $(14, 14)$. Other training settings follow those in DeiT [16] and SwinTransformer [5], except the stochastic depth regularization, which is not used in our experiments. Each student is trained for 300 epochs with 8 Tesla-V100 GPUs.

**Results.** Table 2 presents classification results on ImageNet-1k. When compared with the hard logits distillation proposed in DeiT [16], our manifold method achieves remarkable performance improvements. For example, Deit-Tiny distilled via manifold method outperforms Deit by $+2.0\%$ top-1 accuracy, and Deit-Small outperforms Deit by $+0.9\%$ with the CaiT-S24 teacher. Although the "soft logits distillation" can boost the result by 0.4% compared to the "hard logits distillation", our proposed manifold still obtains better performance by $+0.6\%$. When the teacher is CaiT-XXS24, a much weaker architecture, the corresponding improvements are $+1.6\%$ on DeiT-Tiny and $+1.2\%$ on DeiT-Small, respectively. Note that we report the RegNetY-16GF teacher results in DeiT for comparison, but do not conduct fine-grained manifold distillation experiments with this CNN model because the proposed method is designed for distillation between vision transformers.

---

[2]Mindspore: https://mindspore.cn/resources/hub. Pytorch code: https://github.com/Hao840/manifold-distillation and https://github.com/huawei-noah/Efficient-Computing.

Table 2: Distillation results on ImageNet-1k with 224×224 input. In the first column, "Hard" indicates the hard-label distillation strategy, "Soft" indicates the soft-label based KD method.

| Distillation method | Teacher | Top-1(%) | Student | Top-1(%) |
|---|---|---|---|---|
| - | - | - | DeiT-Tiny | 72.2 |
| Hard [16] | RegNetY-16GF | 82.9 | DeiT-Tiny | 74.5 |
| Hard [16] | CaiT-XXS24 | 78.5 | DeiT-Tiny | 73.9 |
| Hard [16] | CaiT-S24 | 83.4 | DeiT-Tiny | 74.5 |
| Soft [14] | CaiT-S24 | 83.4 | DeiT-Tiny | 74.9 |
| Manifold (ours) | CaiT-XXS24 | 78.5 | DeiT-Tiny | **75.5** |
| Manifold (ours) | CaiT-S24 | 83.4 | DeiT-Tiny | **76.5** |
| - | - | - | DeiT-Small | 79.9 |
| Hard [16] | RegNetY-16GF | 82.9 | DeiT-Small | 81.2 |
| Hard [16] | CaiT-XXS24 | 78.5 | DeiT-Small | 80.1 |
| Hard [16] | CaiT-S24 | 83.4 | DeiT-Small | 81.3 |
| Manifold (ours) | CaiT-XXS24 | 78.5 | DeiT-Small | **81.3** |
| Manifold (ours) | CaiT-S24 | 83.4 | DeiT-Small | **82.2** |
| - | - | - | Swin-Tiny | 81.2 |
| Soft [14] | Swin-Small | 83.2 | Swin-Tiny | 81.7 |
| Manifold (ours) | Swin-Small | 83.2 | Swin-Tiny | **82.2** |

Table 3: Ablation study of main components. The "✓" mark indicates whether we adopt the corresponding training strategy. The "Hard" logits label is the hard-label distillation scheme in DeiT. We adopt a CaiT-S24 teacher and a DeiT-Tiny student.

| Manifold Distillation | Logits label | Fixed Depth | Top-1(%) |
|---|---|---|---|
| × | Hard | × | 74.5 |
| × | Soft | × | 74.9 |
| × | Hard | ✓ | 75.5 |
| × | Soft | ✓ | 75.8 |
| ✓ | Hard | ✓ | 75.9 |
| ✓ | Soft | ✓ | **76.5** |

Moreover, we conduct experiments based on Swin Transformer, one of the state-of-the-art vision transformer architecture. When the teacher is Swin-Small and the student is Swin-Tiny, our proposed method surpasses the original student by $+1.0\%$ and achieves $+0.5\%$ performance improvement compared with the original soft logits label based KD method, demonstrating the effectiveness of our fine-grained manifold distillation.

### 4.3 Ablation and parameter comparison

**Ablation of main components.** We design experiments to verify the effectiveness of our proposed fine-grained manifold distillation method, the fixed student depth, together with the "hard" and "soft" logits label in previous distillation methods. As shown in Table 3, soft logits label can bring $+0.5\%$ top-1 accuracy compared to hard logits label, and fixed depth can improve the baseline by $+0.8$-$1.0\%$, which is a practical strategy to distill transformer. Moreover, when combined with the fine-grained manifold distillation, student performance can be further improved by $+0.7\%$.

**Layers for fine-grained manifold distillation.** To study the impact of different layer selections in manifold relation map matching, we evaluate 5 layer selecting schemes and compare their performance. In particular, we take CaiT-S24 as the teacher model and DeiT-Tiny as the student model. The number of selected layers is set to 6.

From the results reported in Table 5, conducting fine-grained manifold distillation at the head and the tail of student models are both crucial. Hence, we speculate that the "Shallow/Deep" layer selecting scheme is the best because of its outstand-

Table 4: Ablation study on different selected layer numbers. We denote the indices of layers selected from a $L$-layer model as $\{1, 2, ..., k, L-k+1, ..., L-1, L\}$, and conduct experiments with the CaiT-small teacher ($L = 24$) and the DeiT-Tiny ($L = 12$) student to compare different $k$ setting.

| Num. of selected layers | Top-1(%) |
|---|---|
| 2 ($k$=1) | 76.3 |
| 4 ($k$=2) | 76.4 |
| 8 ($k$=4) | 76.5 |
| 12 ($k$=6) | 76.5 |

Table 5: Distillation results with different layer selecting schemes. We use a CaiT-S24 teacher and a DeiT-Tiny student to perform fine-grained manifold distillation. A total of 6 layers are selected from the "Shallow", "Medium", or "Deep" part of a model. "Uniform" refers to selecting layers across the whole model uniformly. Note that the number of select layers is different from other experiments.

| Scheme | Teacher layers | Student layers | Top-1(%) |
|---|---|---|---|
| Shallow | $\{1, 2, 3, 4, 5, 6\}$ | $\{1, 2, 3, 4, 5, 6\}$ | 75.8 |
| Deep | $\{19, 20, 21, 22, 23, 24\}$ | $\{7, 8, 9, 10, 11, 12\}$ | 75.5 |
| Shallow/Deep | $\{1, 2, 3, 22, 23, 24\}$ | $\{1, 2, 3, 10, 11, 12\}$ | **76.3** |
| Shallow/Medium/Deep | $\{1, 2, 12, 13, 23, 24\}$ | $\{1, 2, 6, 7, 11, 12\}$ | 76.2 |
| Uniform | $\{4, 8, 12, 16, 20, 24\}$ | $\{2, 4, 6, 8, 10, 12\}$ | 76.2 |

Table 6: Ablation study results of decoupled manifold loss terms. The teacher is CaiT-S24 and the student is DeiT-Tiny. $\ddagger$ indicates that we use the full precision training instead of mixed precision training for the result.

| Loss term | | | Top-1(%) |
|---|---|---|---|
| $\mathcal{L}_{intra}$ | $\mathcal{L}_{inter}$ | $\mathcal{L}_{random}$ | |
| $\times$ | $\times$ | $\times$ | 75.8 |
| $\checkmark$ | $\times$ | $\times$ | 76.0 |
| $\times$ | $\checkmark$ | $\times$ | $75.8^{\ddagger}$ |
| $\times$ | $\times$ | $\checkmark$ | 76.2 |
| $\times$ | $\checkmark$ | $\checkmark$ | 76.1 |
| $\checkmark$ | $\times$ | $\checkmark$ | 76.4 |
| $\checkmark$ | $\checkmark$ | $\times$ | 76.0 |
| $\checkmark$ | $\checkmark$ | $\checkmark$ | **76.5** |

Table 7: Comparison of different hyper-parameter settings. The teacher is CaiT-S24 and the student is DeiT-Tiny.

| Hyper-parameter | | | | Top-1(%) |
|---|---|---|---|---|
| $\alpha$ | $\beta$ | $\gamma$ | $K$ | |
| **2.0** | 0.1 | 0.2 | 192 | 76.1 |
| **8.0** | 0.1 | 0.2 | 192 | 76.1 |
| 4.0 | **0.05** | 0.2 | 192 | 76.2 |
| 4.0 | **0.2** | 0.2 | 192 | 76.2 |
| 4.0 | 0.1 | **0.1** | 192 | 76.2 |
| 4.0 | 0.1 | **0.4** | 192 | 76.2 |
| 4.0 | 0.1 | 0.2 | **96** | 76.0 |
| 4.0 | 0.1 | 0.2 | **384** | 76.3 |
| **4.0** | **0.1** | **0.2** | **192** | **76.5** |

ing performance and ease of use. Following this selecting strategy, we further ablate the influence of the number of selected layers. We denote the indices of layers selected from a $L$-layer model as $\{1, 2, ..., k, L - k + 1, ..., L - 1, L\}$, and conduct experiments with the CaiT-small teacher ($L = 24$) and the DeiT-Tiny ($L = 12$) student to compare different $k$ setting. The corresponding are shown in Table 4. We can find that our method is robust with various layer numbers, selecting 4 layers ($k = 2$) obtains 76.4 top-1 accuracy, and selecting 8 ($k = 4$) and 12 ($k = 6$) layers achieve 76.5 top-1 accuracy, respectively.

**Ablation of decoupled manifold loss terms.** The decoupled fine-grained manifold distillation loss consists of three terms. We study their effectiveness and report the results in Table 6. The results show that each component in the decoupled fine-grained manifold distillation loss contributes to improving the final performance. In particular, only transferring inter-image relation maps leads to an unstable training process (mixed precision training will corrupt the student model, leading to a NAN loss), we speculate that these relations are highly related to the randomly sampled image batches. When equipped with intra-image and random-image maps, the inter-image relation maps can further improve the accuracy by +0.1%.

**Parameters in decoupled manifold distillation loss.** There are 4 hyper-parameters in decoupled manifold distillation loss: the weight of inter-image loss $\alpha$, the weight of intra-image loss $\beta$, the weight of randomly sampled loss $\gamma$, and the sampling number $K$. We compare different settings of these parameters and report the results in Table 7. Our default setting outperforms those either increasing or reducing one of the parameters. However, due to our coarse parameter searching scheme, we believe that the performance can be further improved by setting these hyper-parameters more carefully.

## 4.4 Transfer learning

To measure the generalization ability of the proposed method, we conduct transfer learning experiments. We fine-tune a DeiT-Tiny student trained with a CaiT-S24 teacher on CIFAR-10 and CIFAR-100 datasets for 1000 epochs. We adopt a batch size of 768 and an AdamW optimizer with a learning rate of $5 \times 10^{-6}$. Other settings follow DeiT [16].

Table 8: Transfer learning results on CIFAR-10/100. Each student is fine-tuned for 1000 epochs.

| Dataset | Teacher | Student | Distillation | Top-1(%) |
|---------|---------|---------|--------------|----------|
| CIFAR-10 | - | DeiT-Tiny | - | 98.19 |
| | CaiT-S24 | DeiT-Tiny | Hard | 98.23 |
| | CaiT-S24 | DeiT-Tiny | Manifold | **98.48** |
| CIFAR-100 | - | DeiT-Tiny | - | 86.61 |
| | CaiT-S24 | DeiT-Tiny | Hard | 87.34 |
| | CaiT-S24 | DeiT-Tiny | Manifold | **88.05** |

Table 9: Object detection results on COCO 2017. Backbones are pre-trained on ImageNet-1k. The teacher is trained for 36 epochs and each student is trained for 12 epochs.

| Model | #params | FLOPs | $AP^{box}$ | $AP^{box}_{50}$ | $AP^{box}_{75}$ |
|-------|---------|-------|-----------|-----------------|-----------------|
| (Teacher) Swin-S + Mask R-CNN | 69M | 365G | 48.5 | 70.2 | 53.5 |
| (Student) Swin-T + Mask R-CNN | 48M | 272G | 43.7 | 66.6 | 47.7 |
| (Manifold distilled) Swin-T + Mask R-CNN | 48M | 272G | **44.7** | **67.1** | **48.6** |

Table 8 reports the transfer learning results. Students trained with the fine-grained manifold distillation method generalize better than others (+0.25% on CIFAR-10 and +0.71% on CIFAR-100), demonstrating a favorable generalization ability of the proposed method.

### 4.5 Downstream task

To further evaluate the effectiveness of our proposed method, we adopt the fine-grained manifold distillation to train object detection models on COCO 2017 dataset and semantic segmentation models on ADE20K dataset.

**Implementation details on COCO.** We adopt Mask R-CNN [45] as the detection framework. The teacher backbone is Swin-Small and the student backbone is Swin-Tiny. All used backbones are pretrined on ImageNet-1k. The teacher is trained for 36 epochs and each student is trained for 12 epochs. When training students with fine-grained manifold distillation, we adopt the distillation loss on outputs of the last two backbone stages and outputs of the feature pyramid network neck [46].

**Results on COCO.** Table 9 presents the detection results. Our manifold distilled student outperforms the student trained without distillation (+1.0% box AP), demonstrating that the proposed method benefits the object detection downstream task.

**Implementation details on ADE20K.** We adopt UPerNet [47] as the segmentation framework. The student is trained for 160K iterations following [5]. Other settings follow experiments on COCO.

**Results on ADE20K.** Table 10 presents the segmentation results. The proposed method outperforms the ditect training

Table 10: Semantic segmentation results on ADE20K. The student is trained for 160K iterations.

| Model | #params | FLOPs | mIoU |
|-------|---------|-------|------|
| (Teacher) Swin-S + UPerNet | 81M | 1038G | 47.64 |
| (Student) Swin-T + UPerNet | 60M | 945G | 44.51 |
| (FitNet distilled) Swin-T + UPerNet | 60M | 945G | 44.85 |
| (Manifold distilled) Swin-T + UPerNet | 60M | 945G | **45.66** |

method and FitNet [48], an intermediate feature distilling approach, indicating that the manifold distillation approach can also help train a semantic segmentation model.

## 5 Conclusion

This paper proposes a fine-grained manifold distillation method for vision transformers. We match patch and batch level features of student and teacher in a manifold space, and decouple the matching loss into three terms to reduce the computational complexity. Moreover, we adopt a patch merging scheme to further simplify the computation. Different from previous works, the student is distilled with soft labels and fixed depth. Experiments are conducted on ImageNet-1k, CIFAR-10/100, MS COCO, and ADE20K. Corresponding results demonstrate that our manifold KD consistently outperforms existing methods. The large search space of hyper-parameters is one of the most serious drawbacks of the proposed method. In the future, we will study to further simplify the proposed method and help it be more parameter insensitive.

**Acknowledgment.** This work is supported by National Key Research and Development Program of China under No. 2021YFC3300200. Also, this work is partially supported by the National Nature Science Foundation of China under Grant 62071013 and 61671027, and National Key R&D Program

of China under Grant 2018AAA0100300. We gratefully acknowledge the support of MindSpore, CANN (Compute Architecture for Neural Networks) and Ascend AI Processor used for this research.

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
