# OpenReview forum: "Learning Efficient Vision Transformers via Fine-Grained Manifold Distillation"
_NeurIPS.cc/2022/Conference — NeurIPS 2022 Accept_

### Official Review · Reviewer_eVwU · 2022-07-06

**Rating:** 5
**Confidence:** 4
**Soundness:** 2 fair
**Presentation:** 2 fair
**Contribution:** 2 fair

**Summary:**

This paper presents a fine-grained patch-level distillation method for vision transformers. Specifically, the proposed method matches the student and teacher models in the patch-level manifold space. The proposed method further decouples the relation map into three terms to solve the high computational complexity brought by the patch-level distillation. Experimental results on various vision tasks and benchmarks demonstrate the effectiveness of the proposed method.

**Questions:**

1. Some notations are confusing. For example, in Section 3.3, the authors use the same notation $F$ for the feature maps before and after reshaping/merging operations. Please make the notations more distinguishable.

2. In Table 2, for different students, the authors only compare the proposed method with either the “Hard” or “KD”. It would be better for the authors to compare the proposed method with both “Hard” and “KD”, which would strengthen the paper.

3. The authors manually select teacher-student layers as distillation pairs. However, it is unclear how the number of selected layers would affect the performance. Please provide more discussions and experimental results.

4. The authors introduce a pair of hyper-parameters $(H^{\prime}, W^{\prime})$ for patch merging. However, the effect of different values of $(H^{\prime}, W^{\prime})$ is unclear. It would be better for the authors to provide more discussions and experiments, which would strengthen the paper.

5. In Table 5, the authors state that training with only inter-image patch-level distillation loss fails because of a nan loss value. In this case, how to obtain the Top-1 accuracy? Why does the model fail to converge? It would be better for the authors to provide more explanations.

6. Minor issues

(1) In Line 3 of Page 1, “... their ...” should be “... them ...”.

(2) In Line 42 of Page 2, “... result ...” should be “... results ...”.

(3) In Line 84 of Page 3, “train ...” should be “trained ...”.

(4) In Line 123, the notations for features of the student and teacher layer should be $F_S$ and $F_T$, respectively.

(5) In Line 143, the dimension of $F_r$ should be $K \times D$.

(6) In Line 172 of Page 6, “... that and a ...” should be “... that a ...”.

(7) In Line 198 of Page 6, “... model.” should be “... models.”.

(8) In Line 208 of Page 7, “... grad ...” should be “... grid ...”.

(9) In Line 228 of Page 8, “... demonstrates ...” should be “... demonstrate ...”.

(10) In Line 232 of Page 8, “... result ...” should be “... results ...”.

(11) In Page 8, the reference position of Tables 5 and 6 seems reversed.

**Limitations:**

The authors do not provide the potential negative societal impact of their work.

**Strengths And Weaknesses:**

Strengths:
1. The authors propose a manifold learning-based knowledge distillation method for vision transformers to match patch-level manifold information between teacher and student models.

2. The authors decouple the relation map into three terms, reducing the computational complexity in computing the patch-level matching loss.

3. Extensive experiments demonstrate the effectiveness of the proposed method.

Weaknesses:
1. The contribution of the paper is limited. The proposed soft distillation has been proposed in [1]. Relying on the proposed patch-level distillation loss, the authors simply combine [1] to learn a compact vision transformer. Moreover, the proposed fixed depth seems to be an empirical trick to adjust the regularization strength.

2. The proposed method belongs to the relation-based knowledge distillation [2]. What is the difference between the proposed method and existing relation-based knowledge distillation methods [3][4]? It would be better for the authors to provide more discussions.

Reference:
[1] Distilling the knowledge in a neural network. arXiv 2015.
[2] Knowledge Distillation: A Survey. IJCV 2021.
[3] A gift from knowledge distillation: Fast optimization, network minimization and transfer learning. CVPR 2017.
[4] Self-supervised knowledge distillation using singular value decomposition. ECCV 2018.

---

> ### Author Response · Authors · 2022-08-02
> **Response to Reviewer eVwU (Part 1/2)**
>
> Thanks for your valuable comments, we have updated the rebuttal revision accordingly.
>
> > **Weakness 1:** The contribution of the paper is limited. The proposed soft distillation has been proposed in [1]. Relying on the proposed patch-level distillation loss, the authors simply combine [1] to learn a compact vision transformer. Moreover, the proposed fixed depth seems to be an empirical trick to adjust the regularization strength.
>
> **Response to weakness 1:** Soft distillation has been proposed for several years. Our main contribution is not the soft distillation, but are about the following two aspects. (i) We propose to leverage both batch and patch level information for knowledge distillation. Given input samples within a mini-batch, all patch embeddings depict the sturcture of a potential manifold space. We train the student to project input into a similar manifold space as the teacher. (ii) We decouple the manifold map to simplify the computation. Since the structure of manifold space is described by relations among all embeddings, the computational cost is usually unaffordable. To reduce the computational burden, we orthogonally decouple the manifold relations into intra-image relations and inter-image relations, and introduce relations among a random sampled subset of embeddings as an auxiliary manner to capture both batch and patch level information.
>
> As for the fixed depth setting, we acknowledgement that it is an empirical trick and it is not our technical contribution. It can bring new insight for the research community: the regularization schemes should be carefully designed for knowledge distillation task, while directly adopting the training skills working well for teacher models may bot be the optimal choices.
>
> > **Weakness 2:** The proposed method belongs to the relation-based knowledge distillation [2]. What is the difference between the proposed method and existing relation-based knowledge distillation methods [3] [4]? It would be better for the authors to provide more discussions.
>
> **Response to weakness 2:** Our method is quite different from the mentioned reference [3] and [4]. [3] assumes the input and output shape of a model layer are $F_{in}\in R^{H\times W\times M}$ and $F_{out}\in R^{H\times W\times N}$, the transferred knowledge matrix of shape $N\times M$ is the product of reshaped $F_{in}$ and $F_{out}$, where $H$, $W$, and $M/N$ are the height, width, and channel dimension of features, respectively. In reference [4], the authors process knowledge in a similar way, except the use of a singular value decomposition (SVD) based feature compressing scheme and a radial basis function (RBF) based matching scheme. These two works focus on transferring the relation between different layers of a model, which describes the structure of a information flow in a model. However, out method targets at transferring the structure of the manifold feature space. Moreover, our proposed manifold relation maps could transfer more generalized batch-information and fine-grained patch-level information, which is impracticable with the two mentioned references. We will add these discussions and the referenced papers to our revision paper.
>
>
> **Reference:**
>
> [1] Touvron, Hugo, et al. "Training data-efficient image transformers & distillation through attention." *International Conference on Machine Learning* (2021).

---

> > ### Author Response · Authors · 2022-08-02
> > **Response to Reviewer eVwU (Part 2/2)**
> >
> > **Response to Q1:** Many thanks for your valuable suggestion, we replace the confusing notation in the revised paper.
> >
> > **Response to Q2:** Thanks for your suggestion. In the original design of vision transformers, outputs of a classification token is used as extracted features. The "Hard" method introduces an additional distillation token to learn from a CNN teacher, and uses averaged outputs of the two tokens as features. However, Swin Transformer model family adopts an average pooling operation on all image tokens to extract features, indicating that the adding of an additional distillation token is not allowed. Hence, we cannot use the "Hard" method to train a Swin Transformer student, and can only adopt the "KD" method in turn. For DeiT students, we design a simple experiment with a CaiT-small teacher and a DeiT-Tiny studnet to evaluate the "KD" method, and corresponding results are as follows:
> >
> > |  method   | Top-1(%) |
> > | :-------: | :------: |
> > | KD (soft) |  $74.9$  |
> > |   Hard    |  $74.5$  |
> > | Manifold  |  $76.5$  |
> >
> > The results shows that "KD" method outperforms the "Hard" method, indicating that "KD" is a stronger baseline than "Hard" when both the teacher and the studnet are vision transformers. To strengthen our paper, We will complete all experiments of training DeiT students with the "KD" method and update the results in our final submission.
> >
> > **Response to Q3:** Thanks for your suggestion. Table 4 shows that selecting layers at the shallow/deep part of a model obtains the best result. Following this selecting strategy, we ablate the influence of the number of selected layers. We denote the indices of layers selected from a $L$-layer model as {$1, 2, ...,k,L-k+1,...,L-1, L$}, and conduct experiments with the CaiT-small teacher ($L=24$) and the DeiT-Tiny ($L=12$) student to compare different $k$ setting. The results are listed as follows:
> >
> > | $k$  | Top-1(%) |
> > | :--: | :------: |
> > | $1$  |  $76.3$  |
> > | $2$  |  $76.4$  |
> > | $4$  |  $76.5$  |
> > | $6$  |  $76.5$  |
> >
> > We can find that our method is robust with various layer numbers, selecting 4 layers ($k=2$) obtains 76.4 top-1 accuracy, and selecting 8 ($k=4$) and 12 ($k=6$) layers achieve 76.5 top-1 accuracy, respectively.
> >
> > **Response to Q4:** Thanks for pointing out this. We will add more discussions in our final version. In our experiments, the patch mergting scheme is only adopted to Swin Transformer students, whose extremely small patch size setting results in an unaffordable GPU memory usage. By default, Swin Transformers use a patch size of $p=4$. When the input resolution is $224\times 224$, there are $56\times 56$ patches in the first model stage. Following the patch number in standard vision transformers is $14\times 14$, we adopt a patch merging setting of $(H'=14,W'=14)$, indicating that every $4\times 4$ adjacent patches are merged. For comparison, we conduct experiments with other two merging settings and report the corresponding results as follows:
> >
> > | setting of $(H',W')$ | size of merged patch | Top-1(%) |
> > | :------------------: | :------------------: | :------: |
> > |      $(28,28)$       |     $2\times 2$      | $82.27$  |
> > |      $(14,14)$       |     $4\times 4$      | $82.24$  |
> > |       $(7,7)$        |     $8\times 8$      | $82.12$  |
> >
> > We can find that increasing or decreasing the patch meaging setting $(H',W')$ has little effect on the Swin Transformer student performance. However, if the merged patch number is increased excessively, following our response to the first weakness proposed by reviewer *ipbi*, it greatly damages the student accuracy. We further conduct this experiment on a DeiT-Tiny student with a patch merging setting of  $(H'=2,W'=2)$ here:
> >
> > | setting of $(H',W')$ | merge every | Top-1(%) |
> > | :------------------: | :---------: | :------: |
> > |       $(2,2)$        | $7\times 7$ |  $74.8$  |
> > |         None         |    None     |  $76.5$  |
> >
> > **Response to Q5:** When we train the student with the failed setting, a NAN loss occurs at the 126th training epoch (300 in total). Hence, we report the Top-1 accuracy of the model saved at the epoch of 125. When preparing this rebuttal, we speculate that the mixed precision training leads to the failure, and adopt a full precision setting to re-run the experiment, where the training converges and the student achieves $75.8\%$ Top-1 accuracy. In particular, we think that only transferring inter-image relation maps leads to an unstable training process, since these relations are highly related to randomly sampled image batches. However, mixed precision training cannot handle such an unstable training, and finally corrupts the student model, leading to the NAN loss. We will report the new result in our revised paper.
> >
> > **Response to Q6:** Thank you very much for carefully reading our submission and pointing out the typos. We have revised them in our rebuttal revision.
> >
> > **Response to the limitation:** We will add the societal impact discussion in our final submission.

---

> > > ### Comment · Reviewer_eVwU · 2022-08-09
> > > **Thank you for the response**
> > >
> > > Thanks for the clarifications. I am satisfied with the responses and raise my score.

---

> > > > ### Author Response · Authors · 2022-08-09
> > > > **Response to Reviewer eVwU**
> > > >
> > > > Dear Reviewer eVwU,
> > > >
> > > >
> > > > Thanks again for your constructive comments which will help improve the quality of our draft.
> > > >
> > > >
> > > > Best,
> > > >
> > > > Paper3797 Authors.

---

### Official Review · Reviewer_JpXV · 2022-07-09

**Rating:** 7
**Confidence:** 3
**Soundness:** 3 good
**Presentation:** 3 good
**Contribution:** 3 good

**Summary:**

The authors found the existing vanilla distillation in ViT can not work with different embedding dimensions between the teacher and student model. Moreover, the authors argue previous ViT distillation process is conducted at a coarse level.
To resolve the above issues, they propose a fine-grained patch-level manifold distillation method.
By decomposing the distillation process into inter and intra parts, the authors found the results have better accuracy with negligible computational overhead.

**Questions:**

1. Why does the Soft Distillation (3rd row) in Table 3 improve only 0.3% compared with the 2nd row?
2. The proposed manifold-based KD still belongs to a subset of KD, but why the authors separate the proposed MD from the original KD, in the caption of Table 2?
3. I appreciate the authors admit the failure of distillation with L_inter, but why do not you re-run the experiment and report a new number?
4. Why missing semantic segmentation experiments as an extension of the downstream tasks for the proposed new KD?

**Ethics Review Area:**

["I don’t know"]

**Limitations:**

The authors should add discussions about societal impact.
For other limitations, please refer the question part.

**Strengths And Weaknesses:**

1. originality:
Manifold learning-based KD, which does not require matching dimensions among teacher and student networks, was first implemented in ViTs.
To reduce the computational complexity, the relation maps are decomposed into three parts for efficiency.
2. quality:
The paper is well written, with detailed experiments and ablation studies with other methods and the proposed variants.
3. clarity:
The paper consists of an illustration of the decoupled manifold relation map, text explanations of the proposed methods.
4. significance:
The paper solves the problem that distillation of different hidden dimensions in ViTs, reduces computational cost than its baseline, and improves the distilled accuracy of previous ViT+distillation methods.

---

> ### Author Response · Authors · 2022-08-02
> **Response to Reviewer JpXV**
>
> Thanks for your valuable comments, we have updated the rebuttal revision accordingly.
>
> > **Question 1:** Why does the Soft Distillation (3rd row) in Table 3 improve only 0.3% compared with the 2nd row?
>
> **Response to question 1:** We speculate that there may be some misunderstanding about notations of Table 3, and we will refine the expression to clarify them in our revision paper. Our baseline, the first line with three $\times$ marks, is the method proposed in [1], which uses hard labels from the teacher to perform knowledge distillation. The result in the original 2nd line with "Soft distillation $\times$" does not indicate that no distillation scheme is used, but a hard logits label based knowledge distillation method is adopted. The $0.3\%$ improvement is achieved by comparing with the hard logits label distillation method. We update the notation from "Soft distillation" to "Logits label" in Table 3, and we further design two ablation experiments to evaluate the impact of soft logits label (2nd & 5th line):
>
> | Manifold Distillation | Logits label | Fixed Depth  | Top-1(%) |
> | :-------------------: | :----------: | :----------: | :------: |
> |       $\times$        |     Hard     |   $\times$   |  $74.5$  |
> |       $\times$        |     Soft     |   $\times$   |  $75.0$  |
> |       $\times$        |     Hard     | $\checkmark$ |  $75.5$  |
> |       $\times$        |     Soft     | $\checkmark$ |  $75.8$  |
> |     $\checkmark$      |     Hard     | $\checkmark$ |  $75.9$  |
> |     $\checkmark$      |     Soft     | $\checkmark$ |  $76.5$  |
>
> The results in this new 2nd line shows that soft logits itself brings $0.5\%$ accuracy improviment compared with hard logits. Moreover, when working together with other modules, soft logits still helps improve the student accuracy.
>
> > **Question 2:** The proposed manifold-based KD still belongs to a subset of KD, but why the authors separate the proposed MD from the original KD, in the caption of Table 2?
>
> **Response to question 2:** Thanks for the valuable comments. The separateion is just for clear notations. The proposed method surely belongs to a subset of KD. Actually, the method notated by "Hard" also belongs to this subset. The sign "KD" in Table 2 is not the generalized mean of knowledge distillation, i.e., the model compression task, but refers in particular to the original distillation method proposed by Hinton *et al.* [2]. We will update the "KD" to "Soft" to eliminate the ambiguity.
>
> > **Question 3:** I appreciate the authors admit the failure of distillation with L_inter, but why do not you re-run the experiment and report a new number?
>
> **Response to question 3:** Thanks for your valuable comments. When preparing our submission, we use the mixed precision training following the DeiT codebase and the experiments fail to converge (NAN at 126 epoch). During rebuttal, we adopt a full precision setting and re-run the experiment, where the NAN loss problem is fixed, and the student achieves $75.9\%$ Top-1 accuracy. We speculate that only transferring inter-image relation maps leads to an unstable training process, since these relations are highly related to random sampled image batches. We will report the new result in our revised paper.
>
> > **Question 4:** Why missing semantic segmentation experiments as an extension of the downstream tasks for the proposed new KD?
>
> **Response to question 4:** Thanks to your valuable suggestion. We have designed experiments on the semantic segmentation task with the ADE20K dataset. The results are as follows:
>
> | model                                  | #params (M) | FLOPs (G) |  mIoU   |
> | :------------------------------------- | :---------: | :-------: | :-----: |
> | (Teacher) Swin-Small+UPerNet           |    $81$     |  $1038$   | $47.64$ |
> | (Student) Swin-Tiny+UPerNet            |    $60$     |   $945$   | $44.51$ |
> | (FitNet Distilled) Swin-Tiny+UPerNet   |    $60$     |   $945$   | $44.85$ |
> | (Manifold Distilled) Swin-Tiny+UPerNet |    $60$     |   $945$   | $45.66$ |
>
> More implementation details, please refer to Section 4.5 in our revised paper.
>
> > **Limitation:** The authors should add discussions about societal impact. For other limitations, please refer the question part.
>
> **Response to the limitation:** Our work does not present any foreseeable societal consequence. We will add the societal impact discussion in our rebuttal revision.
>
> **Reference:**
>
> [1] Touvron, Hugo, et al. "Training data-efficient image transformers & distillation through attention." *International Conference on Machine Learning* (2021).
>
> [2] Hinton, Geoffrey, et al. "Distilling the knowledge in a neural network." *arXiv preprint arXiv:1503.02531* (2015).

---

> > ### Comment · Reviewer_JpXV · 2022-08-07
> > **Response to Authors**
> >
> > Thank you for the detailed response. I appreciate the additional efforts regarding the new experiments and tables, which improve the overall quality of the paper.
> >
> > As a reminder, please include the added experiments on semantic segmentation in the conclusion section.
> >
> > Considering the work is technically sound, and experimentally satisfying, I have increased my initial score.

---

> > > ### Author Response · Authors · 2022-08-08
> > > **Response to Reviewer JpXV**
> > >
> > > Dear Reviewer JpXV,
> > >
> > > Thanks for your support and constructive comments. We have included the added experiments in the conclusion section.
> > >
> > > Best,
> > > Paper3797 Authors.

---

### Official Review · Reviewer_jrGf · 2022-07-11

**Rating:** 6
**Confidence:** 4
**Soundness:** 3 good
**Presentation:** 4 excellent
**Contribution:** 3 good

**Summary:**

This paper works on improving manifold learning-based knowledge distillation when using lightweight vision transformer models. The authors proposed a patch-level fine-grained manifold distillation method by matching the intermediate features between student and teacher in a manifold space and further decoupled the matching loss to reduce the computational complexity.

The proposed approach is evaluated on ImageNet-1K classification task and COCO2017 object detection task and shows better performance than other baselines. Ablation studies are done on the four design components.

**Questions:**

Is there a trade-off between the scheme for reducing computational complexity and the performance on downstream tasks? More explanation and reasoning should be done here.

**Limitations:**

More quantitative evaluation and examples should be provided in the Complexity analysis section.

**Strengths And Weaknesses:**

Strengths: The evaluation and ablation study is relatively comprehensive, covering different downstream tasks and proposed design components.

Weakness: The efficientness of the proposed method is not well supported due to the lack of discussion/details/evaluation on common public benchmarks.

---

> ### Author Response · Authors · 2022-08-02
> **Response to Reviewer jrGf**
>
> Thanks for your valuable comments, we have updated the rebuttal revision accordingly.
>
> > **Weakness:** The efficientness of the proposed method is not well supported due to the lack of discussion/details/evaluation on common public benchmarks.
>
> **Response to the weakness:** Thanks for your advices, we add more details and discussion in Section 4.2 and 4.3, as shown in our "Rebuttal Revision".
>
> > **Questions:** Is there a trade-off between the scheme for reducing computational complexity and the performance on downstream tasks? More explanation and reasoning should be done here.
>
> **Response to the question:** Intuitively, we believe that there must be a trade-off between the computational complexity and the performance on downstream tasks, but the proposed method can greatly reduce the computational complexity with moderate performance degradation. In particular, the original manifold relation map describes relationships among all patches of a mini-batch samples, while the decoupled map only takes a subset into consideration. Such decoupling may lead to an imperfect imitation of the teacher model because of the loss of information. However, this trade-off is inevitable since most common hardwares cannot afford the computation of a complete manifold relation map. Take the object detection task as an example, the input image is of size $1333\times 800$, and the patch size of a Swin-Transformer is $4$ at the first stage, thus there are $334\times 200=66800$ patches in total. The original computation cost and memory usage are quadratic to the number of patches, which makes the computation extremely time-consuming and leads to the out-of-memory problem. To make the computation feasible, we have to decouple the manifold relation map. We speculate that patches in the same image or at the same position are related more closely and provide most of the useful information. Hence, only relation of patches in the same images and patches at the same position of different images is taken into consideration in the proposed scheme. We also randomly select a small set of patches as a complement of information. This scheme can reduce the resource requirement greatly (see our analysis in Section 3.2) while training a well-performed student.
>
> > **Limitation:** More quantitative evaluation and examples should be provided in the Complexity analysis section.
>
> **Response to the limitation:** Thanks for your valuable comments. The original computational complexity and memory usage of a manifold relation map are $\mathcal{O}(B^2N^2D)$ and $B^2N^2$, respectively, where $B$ is the batch size, $N$ is the patch number, and $D$ is the embedding dimension. After decoupling, the two terms becomes $\mathcal{O}(BN^2D+B^2ND+K^2D)$ and $BN^2+NB^2+K^2$, respectively. Ignoring the lower order terms, both the computational complexity and memory usage are reduced by nearly $BN/(B+N)$ times. We will add the quantitative evaluation in the revised paper.

---

### Official Review · Reviewer_ipbi · 2022-07-11

**Rating:** 5
**Confidence:** 4
**Soundness:** 2 fair
**Presentation:** 3 good
**Contribution:** 2 fair

**Summary:**

The authors apply knowledge distillation on vision transformer to learn a more efficient one, so that it could be  deployed on edge devices like cell phones and smart watches.

Considering the knowledge distillation has been fully investigated in neural networks,  the authors utilize the patch-level information and propose a fine-grained manifold distillation method.

To achieve it, the authors use a kind of manifold learning-based KD method, which supports mismatched embedding dimensions. And to utilize the patch-level information in VIT , the authors propose a fine-grained manifold distillation method. The aim is to teach the student layer to output features having the same patch-level manifold structure as the teacher layer.

And in order to simplify the computation, the authors decouple a manifold relation map into three parts, which are intra-image relation map,  inter-image relation map and  randomly sampled relation map.

Together with the path merging strategy, the computation complexity can be further reduced.

**Questions:**

see weakness

**Limitations:**

Yes.

**Strengths And Weaknesses:**

First of all, the motivation of this paper is clear. The authors want to do knowledge distillation to learn a smaller and more efficient vision transformer. And as it is applied in the vision transformer architecture, the authors consider to preserve the relationships among patches.
So, the afterwards proposed schemes are just reasonable. There are mainly two strategies, both of which are proposed to reduce the computation complexity. One is to use three kinds of maps to replace the manifold relation map. The other is to merge patches.

So the weakness of this paper comes, that it doesn't surprise me a lot. It is just looked like a normal solution. In distillation, the manifold map is like a kind of structure information in the pixel-level tasks.

To be more specific, I wonder:
1. If we only use the patch merging strategy, it could reduce the computation complexity a lot. So what is the necessarily to uncouple the map?
2. And when uncoupling the map, what is the exact meaning of the random sampled relation map. I think that the first two kinds may work.
3. And in equation 10, there introduces three weights for balancing each item. When it is applied in each block, I think choosing the weights for each block is a complex task.
4. In line 120-121,  the authors say the teacher-student layers are manually selected, is it reliable enough?

---

> ### Author Response · Authors · 2022-08-02
> **Response to Reviewer ipbi (Part 1/2)**
>
> Thanks for your valuable comments, we have updated the rebuttal revision accordingly.
>
> > **Overall comment:** the manifold map is like a kind of structure information in the pixel-level tasks.
>
> **Response to the overall comment:** We acknowledge that the manifold map is similar to a kind of structure information to some extent, and distilling structural information has always been an effective method in structured prediction problems such as detection and segmentation. However, our manifold map is different from conventional KD solutions for dense prediction in the following two aspects: (i) Utilizing information inside both batch and patch level. Previous methods only consider the structal information in a patch-level-like way, but ignore the information in batch-level. For example, [1] proposes a "pair-wise" distillation to help student preserve the affinity relation in a local region of size $\frac{H'\times W'}{\beta} \times \alpha$, and [2] proposes to distill on a $HW \times HW$ pixel-level relation map. Our propsoed manifold map can fully exploit the information inside bath and patch simultaneously, which is able to transfer the teacher knowledge with a higher fidelity. (ii) Decoupleing the manifold space to simplify computational complexity. Directly leveraging previous "pixel-wise" distillation method for each patch from transformer brings a huge computational burden.Therefore we propose to use three decoupled terms to describe the manifold space and achieve satisfactory result, which is rarely explored in previous structural infromation-based KD.
>
> In addition, we present some practical tips for distilling tranformer-based architecture such as patch merging, fixed-depth, and layer slecting schemes.
>
> **Reference:**
>
> [1] Liu, Yifan, et al. "Structured knowledge distillation for dense prediction." *IEEE transactions on pattern analysis and machine intelligence* (2020).
>
> [2] Shan, Yuhu. "Distilling pixel-wise feature similarities for semantic segmentation." *arXiv preprint arXiv:1910.14226* (2019).

---

> > ### Author Response · Authors · 2022-08-02
> > **Response to Reviewer ipbi (Part 2/2)**
> >
> > **Response to weakness 1:** Only using the patch merging strategy can do reduce the computation complexity, but is not effective and will lead to a worse top-1 accuray. Following the analysis in our paper, the computational complexity of the original manifold map is $\mathcal{O}(B^2N^2D)$, where $N=H\times W$ is the total number of image patches. If we divide the intermediate fature into $\frac{H}{n}\times \frac{W}{n}$ non-overlapping patches, and merge each patch of size $n\times n$ , the complexity becomes $\mathcal{O}(\frac{B^2N^2D}{n^2})$. In our paper, we analyze the complexity on DeiT model, and show that the decoupling method approximately reduces the computational complexity by a factor of $80\times$. To achieve a similar compression rate, patch merging window-size $n$ should be set to $9$, whcih means that there will be only $4$ remaining patches after merging an image with $196$ patches. However, since we aims to utilize the fine-grained patch-level information, such an aggresive merging setting greatly ruined the fine-grained information. We compare the decoupling method and the patch merging method with a DeiT-Tiny student, and the results are listed below:
> >
> > | method                    | FLOPs of a single manifold map | Top-1(%) |
> > | :------------------------ | :----------------------------: | :------: |
> > | only patch merging, $n=7$ |             $4.9$G             |  $74.8$  |
> > | only decoupling           |             $3.0$​G             |  $76.5$  |
> >
> > (We set $n=7$ here since $\sqrt{196}=14$.)
> >
> > In our paper, the manifold map decoupling is to reduce the complexity for capturing information in both batch and patch level, and it is an intuitive implementation to capture the relations between patches in the same image or patches at the same position of multiple images. The patch merging is just an auxiliary strategy to deal with cases where the patch number is too large, e.g., the Swin Transformer, which leads to an unaffordable computation even after decoupling.
> >
> > **Response to weakness 2:** The random sampled relation map serves as complementary information in batch and patch level. Excatly, the first two decoupled maps are enough for knoweldge transfer, but these two maps still ignore relations across patches in different images and different positions. Although the ignored relations are less important intuitively, directly discarding all of them will cause performance drop. To strike the trade-off between the student performance and the computational cost, we randomly sample a subset of these relations to provide more structual information and imbue the student with this relation map. In Tabel 5, the results prove the effectiveness of the randomly sampled relation map.
> >
> > **Response to weakness 3:** The proposed method is robust with various hyper-parameters. In our experiments, we excute an empirical study of hyper-parameters with a CaiT-Small teacher and a DeiT-Tiny student. We compare different hyper-parameter settings and provide the results in Table 6, where the results fluctuate moderately even when we enlarge or shrink one of the weights by two times. Then we adopt the same weight setting for all teacher-student pairs, and every student outperforms its counterpart trained with the vanilla KD method. In practice, users can directly adopt our empirical hyper-parameter setting, and this will guarantee a decent performance.
> >
> > **Response to weakness 4:** We carefully ablate the influences of different layer selecting schemes in Table 4, and we find that selecting layers from both the shallow and the deep part of a model achieves the best result. Therefore, given a model with $L$ layers in total, we can select layers at the shallow/deep model indexed by {$1, 2, ...,k,L-k+1,...,L-1, L$}. And this shceme is general and can be easily applied to any teacher-student model, only if the teacher and the student adopt the same $l$ setting. Take $k=3$ as an example, we select layer {1,2,3,22,23,24} from CaiT-small teacher ($L=24$) and layer {1,2,3,10,11,12} from DeiT-Tiny ($L=12$) student.
> >
> > To evaluate the reliability of the proposed selecting scheme, we further conduct the ablation study on Swin-Small teahcer and Swin-Tiny student, the corresponding results are as follows:
> >
> > |    Scheme    |      Teacher layers      |    Student layers     | Top-1(%) |
> > | :----------: | :----------------------: | :-------------------: | :------: |
> > |   Shallow    |    {1, 2, 3, 4, 5, 6}    |  {1, 2, 3, 4, 5, 6}   |  $81.9$  |
> > |     Deep     | {19, 20, 21, 22, 23, 24} | {7, 8, 9, 10, 11, 12} |  $81.8$  |
> > |   Uniform    |    {4,8,12,16,20,24}     |    {2,4,6,8,10,12}    |  $82.1$  |
> > | Shallow/Deep |     {1,2,3,22,23,24}     |   {1,2,3,10,11,12}    |  $82.2$  |
> >
> > We will add these results to the supplementary material due to the limited pages.

---

> ### Author Response · Authors · 2022-08-09
> **Response to Reviewer ipbi**
>
> Dear Reviewer ipbi:
>
> We thank you for your time and the detailed and valuable reviews!
>
> We would like to kindly remind you that the author-reviewer discussion period ends on Aug 09 ' 22 08:00 PM UTC. We have provided detailed replies and new experiments to your comments. We hope to have a further discussion with you to see if our response solves the concerns. Your support is very important to us and we greatly appreciate that.
>
> Best,
> Paper3797 Authors

---

> ### Comment · Reviewer_ipbi · 2022-08-09
> **Response to authors**
>
> Thanks for the reviewers' rebuttal, which has solved most of my concerns.
>
> So I would like to raise my rating from Borderline reject (4) to Borderline accept (6)
>
> Best,

---

> > ### Author Response · Authors · 2022-08-09
> > **Response to Reviewer ipbi**
> >
> > Dear Reviewer ipbi:
> >
> >
> > Thanks for your feedback!
> >
> >
> > Best,
> >
> > Paper3797 Authors

---

### Meta-Review · Area_Chair_CZtR · 2022-08-26

**Recommendation:** Accept
**Confidence:** Certain

**Metareview:**

Four experts in the field reviewed the paper and recommended Borderline Accept, Weak Accept, Accept, and Borderline Accept. The reviewers generally liked the approach, though some commented that it is straightforward. The reviewers' questions about experiments and clarifications were well addressed by the rebuttal. Hence, the decision is to recommend the paper for acceptance. We encourage the authors to consider the reviewers' comments and make the necessary changes to the best of their ability. We congratulate the authors on the acceptance of their paper!

**Award:**

No

---

### Decision · Program_Chairs · 2022-09-14

Accept